# Neuroinflammatory Response to TNFα and IL1β Cytokines Is Accompanied by an Increase in Glycolysis in Human Astrocytes In Vitro

**DOI:** 10.3390/ijms22084065

**Published:** 2021-04-14

**Authors:** David Pamies, Chiara Sartori, Domitille Schvartz, Víctor González-Ruiz, Luc Pellerin, Carolina Nunes, Denise Tavel, Vanille Maillard, Julien Boccard, Serge Rudaz, Jean-Charles Sanchez, Marie-Gabrielle Zurich

**Affiliations:** 1Department of Biomedical Sciences, University of Lausanne, CH-1005 Lausanne, Switzerland; david.pamies@unil.ch (D.P.); chy.sartori@gmail.com (C.S.); Luc.Pellerin@unil.ch (L.P.); carolina.nunes@unil.ch (C.N.); denisetavel57@gmail.com (D.T.); JustineVanille.Maillard@unil.ch (V.M.); 2Swiss Centre for Applied Human Toxicology (SCAHT), 4055 Basel, Switzerland; Domitille.Schvartz@unige.ch (D.S.); victor.gonzalez@unige.ch (V.G.-R.); julien.boccard@unige.ch (J.B.); Serge.Rudaz@unige.ch (S.R.); Jean-Charles.Sanchez@unige.ch (J.-C.S.); 3Translational Biomarker Group, Department of Internal Medicine Specialties, University of Geneva, CH-1211 Genève, Switzerland; 4Analytical Sciences, Institute of Pharmaceutical Sciences of Western Switzerland and School of Pharmaceutical Sciences, University of Geneva, CH-1211 Genève, Switzerland; 5INSERM U1082, Faculté de Médecine et de Pharmacie, Université de Poitiers, F-86021 Poitiers, France

**Keywords:** astrocytes, astrogliosis, reactive astrocytes, neuroinflammation, energy metabolism, neuroenergetic

## Abstract

Astrogliosis has been abundantly studied in rodents but relatively poorly in human cells due to limited access to the brain. Astrocytes play important roles in cerebral energy metabolism, and are also key players in neuroinflammation. Astroglial metabolic and inflammatory changes as a function of age have been reported, leading to the hypothesis that mitochondrial metabolism and inflammatory responses are interconnected in supporting a functional switch of astrocytes from neurotrophic to neurotoxic. This study aimed to explore the metabolic changes occurring in astrocytes during their activation. Astrocytes were derived from human ReN cell neural progenitors and characterized. They were activated by exposure to tumor necrosis factor alpha (TNFα) or interleukin 1β (IL1β) for 24 h. Astrocyte reaction and associated energy metabolic changes were assessed by immunostaining, gene expression, proteomics, metabolomics and extracellular flux analyses. ReN-derived astrocytes reactivity was observed by the modifications of genes and proteins linked to inflammation (cytokines, nuclear factor-kappa B (NFκB), signal transducers and activators of transcription (STATs)) and immune pathways (major histocompatibility complex (MHC) class I). Increased NFκB1, NFκB2 and STAT1 expression, together with decreased STAT3 expression, suggest an activation towards the detrimental pathway. Strong modifications of astrocyte cytoskeleton were observed, including a glial fibrillary acidic protein (GFAP) decrease. Astrogliosis was accompanied by changes in energy metabolism characterized by increased glycolysis and lactate release. Increased glycolysis is reported for the first time during human astrocyte activation. Astrocyte activation is strongly tied to energy metabolism, and a possible association between NFκB signaling and/or MHC class I pathway and glycolysis is suggested.

## 1. Introduction

The brain uses a very large amount of energy, accounting for about 20% of the total body’s energy consumption [1]. In the past, glucose was reported as the sole valuable energy substrate for the brain. However, subsequent studies demonstrated that other molecules like ketone bodies or lactate play important roles in brain metabolism [2,3,4,5]. Pellerin and Magistretti described, in 1994 [6], a mechanism, now known as the astrocyte–neuron lactate shuttle (ANLS), that could account for the coupling between synaptic activity and energy delivery. This model suggests that glutamate released by activated neurons stimulates glucose uptake and lactate release by astrocytes. Lactate is then taken up by neurons, via monocarboxylate transporters (MCTs), to support the high energy-demanding activity of neurons [6]. Although some aspects of the ANLS are still controversial [7,8], there is considerable support to this mechanism [9,10].

Astrocytes are the most abundant glial cell type in the brain. Their roles are extremely diverse and essential to the proper functioning of the brain. In the last two decades, their cooperation with neurons has been strongly evidenced. It has been established, for example, that astrocytes play a role in neuron signaling modulation [11]. Moreover, the existence of a tight metabolic interaction with neurons has been demonstrated, in particular in terms of energy metabolism [12], defense against oxidative stress [13], and neurotransmitter reuptake and recycling [14]. Another very important feature of astrocytes is their involvement in the neuroinflammatory process. This process, triggered by various stimuli, including trauma, pathogens and environmental chemicals, aims at neutralizing the pathogens and cleaning the tissue of debris [15]. Although microglia are considered as the main central nervous system (CNS)-intrinsic immune cells in the brain, astrocytes are now emerging as important players in the neuroinflammatory response. However, their role in this process has been less well characterized. Astrocytes are immune-competent cells able to detect brain insult signals and respond to them by secreting cytokines and chemokines, and by activating adaptive immune defense [16].

Astrocytes respond to CNS insults through a process called astrogliosis. This reactive state of astrocytes is regulated in a time- and context-specific manner [17]. Different insults with different levels of severity may give rise to varying degrees of astrogliosis [18,19]. Stimuli triggering astrogliosis may shift astrocyte function in at least two different ways, called beneficial and detrimental pathways. In vivo and in vitro observations have revealed mitogen-activated protein kinase (MAPK), NFκB and/or STAT3 activation as critical common events among distinct astrocyte responses. MAPK and NFκB pathway activation participate in the detrimental response while STAT3 activation is part of the protective response [17,20,21,22,23,24,25].

Mitochondrial dysfunction and altered glucose metabolism are thought to play an important role in the progression of neurodegenerative diseases [26,27,28]. Additionally, chronic neuroinflammation is known to be induced by oxidative stress, and various modifications in astrocyte metabolism and inflammation levels have been recently reported as a function of age. These observations led to the hypothesis of an interconnection between mitochondrial metabolism and inflammatory responses, that would support the age-related functional switch of astrocytes, from neurotrophic to neurotoxic [29].

Numerous studies on changes in energy metabolism occurring in astrocytes have been performed in rodent models. However, very little has been done in human astrocytes. Here, we hypothesized that energy metabolism will change in human astrocytes during the inflammatory process triggered by cytokines. To test this, a human astrocyte cell line was exposed to IL1β and TNFα. The inflammatory response was characterized using immunostaining, gene expression and proteomic and metabolomic profiling. Reactive astrocytes showed upregulation of the detrimental pathway and downregulation of the protective pathway, together with increased glycolysis and upregulation of proteins involved in the major histocompatibility complex (MHC) class I pathway, in comparison with non-reactive astrocytes. These results suggest that during astrogliosis, even if detrimental, astrocytes reinforce their energetic support to neurons by releasing more lactate. Alternatively, increased lactate release and upregulation of the major histocompatibility complex (MHC) class I pathway may contribute to attracting and activating microglia.

## 2. Results

### 2.1. ReN-Derived Astrocytes Take Up Aspartate and Display the Classical Astrocyte Marker Glial Fibrillary Acidic Protein (GFAP)

ReN cells are an immortalized human neural progenitor cell line able to differentiate into neurons and glial cells [30]. As quality control of ReN multipotency, cells were differentiated into astrocytes (Figure 1A) and neurons and characterized. Bright field pictures show the difference in morphology between neurons and astrocytes derived from ReN cells (Figure 1B). Immunostaining showed that astrocytes express glial fibrillary acidic protein (GFAP) while neurons are tubulin beta-III (TUBB3) positive, suggesting the expected differentiation (Figure 1B). The ratio of the differentiation of ReN cells in astrocytes is very close to 100%, whereas this ratio is much lower after the differentiation of ReN cells in neurons (merge in Figure 1B). Furthermore, aspartate uptake (Figure 1C), commonly used as a surrogate for glutamate uptake [31], which is a major functional criterion for an astroglial phenotype, was much higher in astrocytes than in neurons (1.61 vs. 0.41 nmol/10^6^ cell per min, respectively), whereas the ReN cell uptake value (1.03 nmol/10^6^ cell per min) was between astroglial and neuronal values. These results confirm the ability of ReN cells to differentiate into neurons and glial cells, and demonstrate that ReN-derived astrocytes are functionally mature.

### 2.2. ReN-Derived Astrocytes Show an Inflammatory Response after Acute Exposure to TNFα or IL1β

In order to characterize their reactivity, ReN cell-derived astrocytes were exposed for 24 h to proinflammatory cytokines (TNFα 10 and 30 ng/mL, IL1β 10 and 30 ng/mL). Cytotoxicity, immunostaining and gene expression analyses were performed for each condition.

A resazurin assay did not reveal cytotoxicity after 24 h of exposure to any of the treatments (Appendix A). Exposure to both concentrations of TNFα induced a strong increase, in most cases significant, in gene expression of the inflammation markers tumor necrosis factor alpha (*TNFα)*, interleukin 6 (*IL6*), interleukin 1β (*IL1β*), beta-2 microglobulin (*B2M*) and proteasome subunit beta type-8 (*PSMB8*) (Figure 2A), indicating an activation of the inflammatory response in astrocytes. Although the upregulation of the intermediate filament protein GFAP in the course of astrocyte activation has been reported by many authors (for a review, see [32,33]), in our study, GFAP expression was strongly downregulated after exposure to all treatments (Figure 2A). The gene expression of vimentin (*VIM*), another intermediate filament protein, was also strongly decreased (Appendix A). An almost total disappearance of S-100 calcium-binding protein, beta chain (*S100B*) mRNA expression was observed (Appendix A) after all treatments, whereas glutamine synthetase (GS), another specific marker of astrocytes, showed no change in expression in any of the treatments. Finally, Ki67, a marker of proliferation, was significantly upregulated after IL1β but not TNFα exposure (Figure 2A). NFκB1, NFκB2 and STAT1 gene expression was found to be strongly and significantly upregulated by both concentrations of TNFα and IL1β (except for STAT1 after the lowest concentration of IL1β), whereas STAT3 mRNA was downregulated after 10 ng/mL IL1β and did not change in the other conditions (Figure 2B), suggesting that ReN-derived astrocytes are activated towards the detrimental pathway [17].

Immunostaining qualitatively showed a decrease in GFAP and vimentin expression after exposure to TNFα or IL1β, particularly at the highest concentration of IL1β (30 ng/mL) where only few stained processes remain visible. After exposure to TNFα, we observed slight modifications in the astrocyte shape, i.e., cell bodies appeared slightly thinner and processes slightly thicker than in control cultures (Figure 3, first row; higher magnification, third row), whereas after IL1β, cell bodies and processes appeared clearly enlarged at 10 ng/mL, and strongly reduced at 30 ng/mL (Figure 3, first row; higher magnification, third row). Cell bodies of S100B-positive cells appeared thicker after exposure to TNFα at both concentrations and IL1β at 30 ng/mL (Figure 3, second row; higher magnification, fourth row).

Moreover, proteomics, metabolomics and pathway enrichment analysis were used to define specific signatures of reactive astrocytes. A PCA of the combined data from the three LC–MS approaches used in metabolomics (Figure 4A) shows that the highest concentration of both treatments induces a similar metabolic profile, as observed by the overlap of samples of TNFα (30 ng/mL) and IL1β (30 ng/mL) on the score plot (Figure 4A). Groups treated with lower concentrations of both proinflammatory cytokines (10 ng/mL TNFα and 10 ng/mL IL1β) showed a specific behavior and diverged from the common pattern observed at 30 ng/mL (Figure 4A). These results suggest that each treatment exerts a characteristic inflammatory effect at low concentrations, leading to a convergent metabolic pattern at higher concentrations.

MetaCore^TM^ curated databases allowed us to perform a pathway enrichment analysis, based on the list of proteins and metabolites showing a statistically significant differential expression in at least three of the four treatments, defined by the selected thresholds for the comparison of interest, i.e., fold change >1.2 and *p*-value < 0.05. As expected, the most relevant pathways identified by MetaCore^TM^ were related to inflammation (Figure 4B), such as “immune response and inflammasome in inflammatory response”, and to metabolism, i.e., “mitochondrial dysfunction in neurodegenerative diseases” (Figure 4B).

### 2.3. TNFα and IL1β Treatments Activate Slightly Differently Inflammatory Pathways in Astrocytes

In order to assess in detail the inflammatory response of reactive astrocytes, the proteomics analysis of TNFα 30 ng/mL and IL1β 30 ng/mL groups was further evaluated. In the top ten regulated pathways, both treatments enhanced three immune pathways: the “IFN-alpha/beta signaling pathway via JAK/STAT”, the “MHC class I, classical pathway”, and the “antiviral action of interferons pathway”. However, TNFα showed higher –log10 (*p*-value) (Figure 4C) for these three pathways. 

Activation of alpha/beta interferon (IFN-α/β) pathways via JAK (Figure 5A) or via MAP (Figure 5B) constitutes an early host defense mechanism. Upregulation of these pathways was stronger after exposure to TNFα than to IL1β. The top 10 upregulated proteins of the IFN signaling via JAK were interferon-induced GTP-binding protein (MX2), interferon-induced protein with tetratricopeptide repeats 2 and 3 (IFIT2, IFIT3), interferon-stimulated gene 15 (ISG15), human leucocyte antigen-B (HLA-B), guanylate-binding protein 1 (GBP1), STAT1, B2M, transporter 1, ATP-binding cassette subfamily B member (TAP1) and HLA-A (Figure 5A). These proteins were strongly (MX2, IFIT2 and 3, ISG15, HLA-B, GBP1) or moderately (STAT1, B2M, TAP1, HLA-A) upregulated by both concentrations of TNFα, whereas after 30 ng/mL of IL1β, only IFIT3, ISG15, HLA-B and GBP1 were also strongly upregulated, the other proteins being either only moderately upregulated (STAT1, HLA-A) or even strongly downregulated (MX2, IFIT2, B2M, TAP1). IL1β at 10 ng/mL mostly downregulated the proteins of this pathway (Figure 5A). For the signaling via MAP, the general pattern of protein regulation was quite similar, with both concentrations of TNFα and the highest concentration of IL1β inducing mostly upregulation of proteins, and the low concentration of IL1β inducing mostly their downregulation (Figure 5B).

Exposure to TNFα and IL1β induced strong modifications in the major histocompatibility complex (MHC) class I pathway (Figure 5C). However, although TNFα induced the upregulation of many proteins at 10 ng/mL, and more at 30 ng/mL, IL1β 10 ng/mL mostly induced the downregulation of proteins, while after IL1β 30 ng/mL, protein upregulation was mostly observed. The top upregulated proteins after TNFα exposure were HLA-A, HLA-B, TAP1, TAP2, B2M and also some members of the proteasome subunit beta (PSMB) protein family (PSMB1, PSMB8 and PSMB9). After IL1β exposure (30 ng/mL), the top upregulated proteins where also HLA-A, HLA-B, PSMB1 and PSMB8 (Figure 5C), however, contrarily to TNFα, IL1β induced a downregulation of B2M, TAP1, TAP2 and PSMB9. Confirming these results, the gene expression of PSMB8 and B2M was also significantly upregulated after both concentrations of TNFα and after the highest concentration of IL1β (Figure 2A). Taken together, these results show that both treatments were able to induce an inflammatory response in astrocytes, and strongly triggered IFNα/β and MHC class I pathways.

### 2.4. Proinflammatory Cytokines Seem to Trigger the Detrimental Pathway in ReN-Derived Astrocytes

Proteomics results showed the upregulation of signal transducer and activator of transcription 1 (STAT1), diverse proteins of mitogen-activated protein kinase (MAPK) and NFκB families (involved in the detrimental pathway) and a decrease in STAT3 protein (involved in protective astrocyte behavior) (Figure 5A). These results confirm the mRNA results for NFκB1, NFκB2, STAT1 and STAT3 described in Section 2.2 (Figure 2B) and reinforce the suggestion that ReN-derived astrocytes are activated towards the detrimental pathway after exposure to TNFα or IL1β [17].

### 2.5. Activation Increases Glycolysis in Astrocytes

Glucose transporter 1 (GLUT1), also known as solute carrier family 2 and facilitated glucose transporter member 1 (SLC2A1), is the main glucose transporter on astrocytes. 

Exposure to TNFα and IL1β showed significant increases in *GLUT1* gene expression (Figure 6A), except for the lowest concentration of IL1β. Gene expression of *MCT4* (SLC16A3), the monocarboxylate transporter mainly found in astrocytes, was strongly and significantly upregulated in all treatments (Figure 6A). PKM2 catalyzes the final irreversible step in glycolysis and generates pyruvate and ATP. All applied treatments induced a significant increase in *PKM2* gene expression, except for the lowest concentration of IL1β (Figure 6A). Altogether, these data suggest that the inflammatory activation of astrocytes modifies their energy metabolism.

In order to functionally evaluate energy metabolism, several parameters were measured: lactate release, oxygen consumption, acidification of the medium. Furthermore, proteomics and metabolomics profiles were established. Seahorse^TM^ data showed a significant increase in basal glycolysis and a significant decrease in basal respiration and ATP production after 10 and 30 ng/mL TNFα and IL1β (Figure 6B). Simultaneously, lactate release was significantly increased after exposure to these cytokines (Figure 6C). On the other side, proteomics data showed that most of the proteins modified in the glycolytic pathway were upregulated, whereas most of the proteins modified in the oxidative phosphorylation pathway were downregulated by TNFα and IL1β (Figure 6D) at 10 and 30 ng/mL. These results suggest that astrocytes activated by TNFα and IL1β increase their glycolytic rate and reduce their oxidative phosphorylation in order to increase the release of lactate.

## 3. Discussion

Animal models have been of critical importance to decipher the different functions and properties now associated with astrocytes. However, some species differences, such as the astrocyte–neuron ratio, astrocyte morphology and gene expression profile, have led to the hypothesis that the regulation of astrocytic functions may differ between humans and rodents (for a review, see [34]). The difficulty in obtaining human astrocytes has clearly been one of the main reasons for the limited number of studies performed on these cells. Here, increased glycolysis and lactate release were observed in reactive human ReN cell-derived astrocytes, and a link with NFκB and MHC class I pathways is suggested. For this study, ReN cell-derived astrocytes were used for their very high reproducibility and robustness.

Astrogliosis is a complex cellular process involving astrocytes in response to various types of CNS injury [35]. Reactive astrocytes are classically characterized by the increased expression of GFAP. In the present study, proinflammatory cytokines profoundly affected the cytoskeleton and the inflammatory response of human ReN-derived astrocytes. In the absence of cytotoxicity, GFAP, vimentin and S100B expression was downregulated, while the GS mRNA level remained unchanged. This type of unusual astrocyte reactivity has already been reported in a complex rat 3D brain cell culture system after exposure to ochratoxin A, together with a stimulation of brain inflammatory response genes [36]. In human ReN-derived astrocytes, we also found a strong activation of inflammatory pathways, showing that in spite of decreased GFAP, astrocytes were highly reactive. In line with these results, some in vitro studies have shown a decrease in GFAP mRNA levels after TNFα and IL1β treatment [37,38]. Taken together, these results suggest that in the absence of cell death, a GFAP decrease might be considered as a sign of astrocyte reactivity, reflecting one of the many forms of astrogliosis, with the knowledge that this phenomenon is highly heterogeneous. However, it may alternatively be due to some unidentified confounding factors.

In agreement with the abundant literature, in the present study, proinflammatory cytokines induced, besides cytoskeletal changes, the expression of transcription factors and genes implicated in the inflammatory response, i.e., NFκB1, NFκB2, TNFα, IL1β and IL6, and downregulated STAT3 (for a review, see [for review, 17]). Astrogliosis is a heterogeneous spectrum of astrocytic responses that can be grouped either as a protective or detrimental phenotype. The outcome of astrogliosis depends on proteins present on the surface of the astrocytes and in their cytoplasm [17]. Upregulation of proteins such as STAT3, ERα and BDNF [17,39,40,41,42,43] has been linked to a protective profile, while upregulation of proteins such as NFκB, C–C motif chemokine ligand 2 (CCL2), C-X-C motif chemokine 10 (CXCL10) and vascular endothelial growth factor (VEGF) [17,44,45,46,47] has been related to the detrimental profile. A protective phenotype leads to clinical improvement, reduction of astrogliosis, immunosuppression and neuronal survival, while the detrimental phenotype is associated with clinical deterioration, enhanced astrogliosis, immune cell recruitment, oxidative stress and neurodegeneration between others [17]. The upregulation of NFκB without change in STAT3 found in the present study suggests that proinflammatory cytokines initiated detrimental astrocyte reactivity. This is also confirmed by the upregulation of PSMB8, a marker of the harmful A1 type of astrocytes described by Liddelow et al. [48]. 

The metabolic profile expressed by cells upon exposure to proinflammatory cytokines shows the metabolic rewiring taking place in reactive astrocytes. Increased levels of biopterin were found in cells treated with TNFα or IL1β, in accordance with the literature, highlighting the value of such a molecule as a marker of neuroinflammation phenomena [49]. Creatine, taurine and ascorbic acid presented a similar profile. These metabolites are involved in the management of inflammatory responses and oxidative stress by playing a protective role [50,51]. Our results stress the complexity of the astrocytic response, showing markers of the harmful A1 phenotype, together with an increase in the production of protective compounds. Depletion of many amino acids (phenylalanine, norleucine, leucine, tyramine, valine) and dopamine was found in the treated astrocytes, pointing towards biomolecular repurposing with regard to non-activated cells. 

Transcription factors of the NFκB family are considered not only as central regulators of immune and inflammatory responses, but are also directly linked with energy metabolism regulation [52]. A very recent study performed on isolated primary mouse astrocytes suggested that the astrocytic inflammatory response driven by nuclear factor kappa B (NFκB) signaling is dependent on glycolytic metabolism [53]. In our human cells, we have observed an increase in NFκB gene expression after TNFα and IL1β exposure (Figure 2B) that also induced an increase in glycolysis and a reduction in oxidative phosphorylation (Figure 6). These results support, for the first time in human cells, the increase in glycolysis during astrocytic activation, and suggest a possible link between NFκB signaling and glycolysis, similarly to that in mice [53]. 

In the present study, proteomics exhibited the activation of the MHC-I pathway. This is interesting in light of the new roles suggested for astroglial MHC-I molecules in the CNS. Indeed, recently, the overexpression of MHC-I in astrocytes has been shown to activate microglial cells, to decrease the number of parvalbumin-positive neurons and to reduce dendritic spine density in the mouse prefrontal cortex, leading to social and cognitive deficits [54]. It has also been shown that B2M, a molecule of the MHC-I pathway found to be upregulated in our study, is required for the development of astrogliosis in vitro [55]. Furthermore, a recent study indicated that besides playing a role in the immune response, HLA class I antigens, and in particular HLA-B and C, are crucial in the metabolism of melanoma cells, sustaining glycolysis [56]. In light of these published data, it could be hypothesized that the MHC class I pathway is involved in the development of astrogliosis and in the increased glycolysis we observed in this study, and in the potential ability of these human astrocytes to activate microglial cells. 

Despite the fact that like most eukaryotic non-proliferative cells, astrocytes rely on oxidative metabolism for energy production, they exhibit a prominent aerobic glycolysis capacity (for a review, see [57]). Aerobic glycolysis, also called the Warburg effect, consists in the conversion of a large proportion of glucose into lactate regardless of oxygen availability [58]. Increased glucose consumption by mouse primary astrocytes has been reported after exposure to proinflammatory cytokines, without an alteration in lactate release [59,60]. In the present study, proinflammatory cytokines caused an increase in GLUT1 transporter mRNA and protein, suggesting increased glucose uptake, as observed in mouse astrocytes. However, this was coupled to increased basal glycolysis and lactate release in ReN-derived human astrocytes. Furthermore, basal respiration and ATP production went down, suggesting that reactive astrocytes are reinforcing their glycolytic metabolism rather than switching to a more oxidative metabolism. The major function of aerobic glycolysis is thought to be the maintenance of high levels of glycolytic intermediates to support anabolic reactions in cells, explaining the fact that it is selected by proliferating cells throughout nature (for a review, see [61]). In the present study, Ki67, which is widely used as a marker of cell proliferation [62], was significantly upregulated after IL1β but not TNFα exposure, suggesting that proliferation is not the only reason for cells to favor glycolysis in the presence of oxygen. Indeed, the astrocyte–neuron lactate shuttle model proposed by Pellerin and Magistretti [6] describes the importance for neurons of lactate release by astrocytes. Therefore, it can be hypothesized that reactive astrocytes increase their glycolytic metabolism in order to supply neurons with lactate, the energy substrate neurons favor after activation by neurotransmitters and that has been shown to be neuroprotective after cerebral ischemia and excitotoxic insult [63,64]. This could help neurons to cope with the insult that triggered the gliosis.

Alternatively, the increase in MCT4 and lactate release observed here might be deleterious for the neurons, if the reported improved cognitive ability and reduced neuronal apoptosis observed in one Alzheimer’s disease mouse model after injection of siMCT4 [65] is confirmed by further studies. The increased lactate produced by reactive astrocytes may also be preferentially directed to microglia, as suggested by the recently proposed hypothesis of the astrocyte–microglia lactate shuttle (AMLS), an adaptation of the ANLS, which would take place under conditions of neuroinflammatory infectious diseases [66,67]. Finally, there is fast growing evidence showing that glycolysis plays a critical role in the activation of immune cells [68,69] and an increase in glycolytic metabolism has also been reported in inflammatory microglia (for a review, see [70]). It could be the same for the activation of astrocytes. In this context, MCT4 was shown to be required for the development of the inflammatory response in a variety of mouse macrophages [71]. In line with this study, we observed a strong upregulation of MCT4 gene expression in reactive astrocytes.

Taken together, the results of this study show that human ReN-derived astrocytes, activated as expected by TNFα and IL1β, most likely towards a deleterious phenotype as seen by changes in the expression of proteins of inflammatory and immune pathways, increase their glycolytic capacity without increasing oxidative phosphorylation, and as a consequence release more lactate. Whether the increased lactate release would help neurons to cope with the stimulus that triggered gliosis, or would, together with the upregulation of MHC-B, contribute to attracting and activating microglial cells needs to be determined in more complex in vitro systems. In any case, astrocyte activation is strongly tied to energy metabolism. Several aspects of astrocyte reaction are still unclear or even controversial, as described by Escartin et al. [35]. In this context, ReN-derived astrocytes constitute an interesting model to study human astrocyte reactions in vitro. Bearing in mind the heterogeneity of brain astrocyte reactions, this model may be further proposed as a tool for toxicological research. However, although we characterized these astrocytes and showed their functionality, we have to keep in mind that they are produced from a cell line. Since human primary astrocytes are not available, further experiments should be performed in other models, such as the recently available astrocytes derived from human induced pluripotent stem cells, for comparison.

In conclusion, this is the first time that increased glycolysis and lactate release have been reported during human astrocyte activation. A possible association between NFκB signaling and/or the MHC class I pathway and glycolysis is suggested. A better understanding of the mechanisms connecting energy metabolism and astrogliosis may lead to improved control of the neuroinflammatory reactions aggravating brain diseases.

## 4. Materials and Methods

### 4.1. Cell Cultures

ReN cell VM human progenitors (ReN) were purchased from Merck (Darmstadt, Germany). The cells were plated in laminin-coated cell culture flasks and maintained in DMEM/F12 medium (Thermo Fisher Scientific Gibco, Waltham, MA, USA) supplemented with GlutaMax 2 mM (Life Technologies, Bleiswijk, Netherlands), B27 without vitamin A (Thermo Fisher Scientific Gibco, Waltham, MA, USA), heparin 1 U/mL (Sigma-Aldrich, Saint-Louis, MO, USA), basic fibroblast growth factor (bFGF) 20 ng/mL (PeproTech, London, UK) and epidermal growth factor (EGF) 20 ng/mL (PeproTech, London, UK), and kept in an incubator with 5% CO_2_. Medium was renewed three times per week until confluency. For differentiation into astrocytes, cells were plated onto laminin-coated 6-well plates (5 × 10^5^ cells/well) in ReN medium. The day after, ReN medium was replaced with GMEM (Thermo Fisher Scientific Gibco, Waltham, MA, USA) with fetal calf serum (FCS) 10% (Thermo Fisher Scientific Gibco, Waltham, MA, USA). Medium was renewed three times per week.

### 4.2. Treatments

At day in vitro 21, astrocytes derived from ReN cells were exposed to TNFα or IL1β (PeproTech, London, UK, 10 and 30 ng/mL each). Stock solutions were prepared in ultrapure H_2_O. For exposure of the cells, aliquots of the stock solutions were added to fresh medium to reach the final nominal concentrations indicated above.

### 4.3. Cell Viability

After treatment, each well was washed once with 1 mL Dulbecco’s phosphate-buffered saline (DPBS), then incubated with 500 μL of resazurin solution (44 mM) for 1 h at 37 °C. The fluorescent product resorufin was detected at 540 nm excitation and 590 nm emission using a Synergy plate reader (BioTeK, Winooski, VT, USA).

### 4.4. Immunohistochemistry

Six glass coverslips were placed in each well of a 6-well plate. Wells were coated with laminin and 5 × 10^5^ cells were plated per well. Cells were cultured as indicated above. After treatments, cells were fixed for 1 h with paraformaldehyde (PFA) 4%. After fixation, coverslips were transferred to a dry surface and staining was done by adding a drop of the different solutions. Astrocytes were incubated for 1 h in blocking solution consisting of 5% normal goat serum (NGS) in DPBS with 0.4% Triton-X100 (Sigma-Aldrich, Saint-Louis, MO, USA). Samples were then incubated for 48 h at 4 °C with a combination of primary antibodies (Appendix A) diluted 1:200 in DPBS containing 3% NGS and 0.1% Triton-X100. After the 48 h, samples were washed 3 times for 5 min in DPBS by adding and removing a drop of DPBS on the top of the fixed cells. Samples were further incubated for 1 h with secondary antibodies (Appendix A) diluted 1:200 in PBS with 3% NGS at room temperature. Subsequently, cells were washed again 3 times for 5 min each with DPBS, the nuclei were stained with Hoechst 33,342 (1:10,000, Thermo Fisher) for 5 min. Finally, samples were mounted on glass slides by using Immu-mount (Thermo Fisher Shandon, Waltham, MA, USA). The images were taken using a Zeiss LSM 780 GaAsP.

### 4.5. Gene Expression

Total RNA was extracted using an RNeasy kit (Qiagen, Hilden, Germany) according to the manufacturer’s guidance. RNA concentration was determined by spectrophotometry using a NanoDrop ND-1000. Reverse transcription was performed on 1 µg total RNA with the HighCapacity cDNA Reverse Transcription kit (Life Technologies, Carlsbad, CA, USA) on a 2720 Thermo Cycler (Thermo Fisher Scientific Applied Biosystems, Waltham, MA, USA). Real-time PCR analyses were performed using Power SYBR Green (LifeTechnologies, Carlsbad, CA, USA) with primers listed in Appendix A, or Taqman Master Mix with probes referenced in Appendix A, using a 7900 HT thermocycler (Thermo Fisher Scientific Applied Biosystems, Waltham, MA, USA). Each sample was analyzed in triplicate. The ΔΔCt method [72] was used to calculate the relative mRNA expression. Data were accepted at <40 cycles of amplification. Results are expressed as fold change compared to untreated control cultures maintained under normal medium conditions, set at 1 as a baseline. Beta-actin (Actb) was used as a reference gene.

### 4.6. Lactate Release

Lactate was measured by a spectrophotometric method in the medium after centrifugation (10 min, 300× *g*, 4 °C). Briefly, 100 µL medium were added to 100 µL of buffer (glycine-semicarbazide hydrochloride 330 mM, NAD 15 mM and lactate dehydrogenase (LDH) 70 U/mL) in 96-well plates. Plates were incubated 1 h at 37 °C, then cooled down to ambient temperature. Absorbance was read at 340 nm on a Synergy plate reader (BioTeK, Winooski, VT, USA). Results were calculated from a standard curve made with lactate (Sigma-Aldrich, Saint-Louis, MO, USA).

### 4.7. Aspartate Uptake

Aspartate uptake was measured using a radioactive method. Briefly, cell culture medium was removed and replaced by 1 mL fresh medium containing radioactive and non-radioactive aspartate (^3^H-D-aspartate 3 µCi/mL, Anawa; D-aspartate 500 µM, Sigma Aldrich). After 5 min, cells were washed 3 times with cold DPBS, then lysed with 1 mL of lysis solution (NaOH 0.1 M, triton X-100 0.1%). Five hundred microliters were transferred to a liquid scintillation counting vial then 4 mL of scintillation liquid (Flo-Scint, Perkin Elmer, Villebon-sur-Yvette, France) were added. Non-specific uptake was evaluated by performing the same procedure on ice. Radioactivity was measured with a TRI-CARB 2300 tr liquid scintillation counter (Packard Instrument Company, Meriden, CT, USA). The uptake was expressed as nmol/10^6^ cells × min^−1^.

### 4.8. Extracellular Flux Analyzer

The oxygen consumption rate (OCR, pmoles O_2_ consumed/min) and the extracellular acidification rate (ECAR, mpH/min) were determined using the Seahorse XF96 Extracellular Flux Analyzer (Agilent Technologies, North Billerica, MA, USA). Two kits were used. The Agilent Seahorse XF Cell Mito Stress Test Kit (Agilent Technologies, North Billerica, MA, USA) quantitates the OCR of cells to measure parameters related to mitochondrial function. The Seahorse XF Glycolysis Rate Assay Kit (Agilent Technologies, North Billerica, MA, USA) preferentially measures the glycolytic function in cells. ReN cells were counted and seeded (2 × 10^4^ cells/well) in XF96 Seahorse^®^ microplates precoated with poly-D-lysin (ThermoFisher Scientific, Waltham, MA, USA). After 21 days of differentiation (as described above), cells were washed 2 times with XF Assay Medium supplemented with 1 mM pyruvate, 2 mM glutamine and 10 mM glucose (all from Agilent technologies, North Billerica, MA, USA). Then, 100 µL of the same medium were added in each well. The plate was left to equilibrate for 1 h in a CO_2_-free incubator before being transferred to the Seahorse XF96 analyzer. The prehydrated cartridge was filled with the indicated compounds and calibrated for 30 min in the Seahorse Analyzer. All the experiments were performed at 37 °C. The Seahorse XF Report Generator automatically calculated the parameters from wave data that were exported to Excel.

### 4.9. Statistical Analysis

Prism 9.0.1 software (GraphPad Software, San Diego, USA) was used for statistical data analysis and graphic representation for cell viability, gene expression, lactate release and extracellular flux analyses. All values are reported as boxplots with datapoints. Statistical analysis was performed using the non-parametric Kruskal–Wallis test followed by Dunn’s multiple comparisons test. Statistics, actual *p*-values, when significant and the number of samples are reported in the figure legends. Statistical significance in the figures is indicated as: * *p* < 0.05; ** *p* < 0.01; *** *p* < 0.001; **** *p* < 0.0001, ns = *p* > 0.05. For each experiment, cultures were randomly attributed to experimental groups.

### 4.10. Proteomics

#### 4.10.1. Sample Preparation for Proteomic Analysis

Four replicates of each treated sample (A1, A2, TNFα 10 ng/mL, TNFα 30 ng/mL, IL1β 10 ng/mL, IL1β 30 ng/mL) as well as control samples were solubilized in RapiGest SF surfactant 0.1% (Waters) in 0.1 M TEAB. After homogenization steps, solubilized proteins were recovered by centrifugation in the supernatants. 

Reduction of 20 µg of protein per sample was done with tris (2-carboxyethyl) phosphine (TCEP), final concentration of 10 mM, and samples reacted for 30 min at 55 °C. Alkylation was performed with iodoacetamide, added to a final concentration of 40 mM, and samples were incubated for 30 min in the dark at room temperature. Trypsin was added (ratio of 1:25, *w*/*w*), and the digestion was performed overnight at 37 °C. Then, three 11-plex tandem mass tag (Thermo Scientific, Rockford, USA) labelings were performed according to the manufacturer’s instructions (experiment 1:4× A1, 4× A2, 3× controls; experiment 2:4× TNFα 10 ng/mL, TNFα 30 ng/mL, 3× controls; experiment 3:4× IL1β 10 ng/mL, 4× IL1β 30 ng/mL; 3× controls). The selected design integrates identical control samples in each experiment as a reference channel, enabling global quantitative analysis. Tandem mass tag reagents were dissolved in acetonitrile (ACN), and each sample was incubated for 60 min at room temperature with a specific tag. For tag quenching, 8 µL of hydroxylamin 5% (*v*/*v*) were added and incubated with the samples for 15 min. After Rapigest cleavage with trifluoroacetic acid (TFA), labeled samples were pooled and dried under vacuum. The samples were dissolved in 5% ACN/0.1% FA and desalted with C18 micro-spin columns (Thermo Scientific, San Jose, CA, USA). Peptides were separated by off-gel electrophoresis [73], and desalted and solubilized in an appropriate amount of 5% ACN/0.1% FA for mass spectrometry (MS) analysis.

#### 4.10.2. MS Data Acquisition for Proteomics Analyses

For LC–MS/MS, peptides were dissolved in 5% ACN/0.1% FA to a concentration of 0.25 µg/µL. Mass spectrometry experiments were performed on a Q Exactive Plus instrument (Thermo Scientific, San Jose, CA, USA) equipped with an Easy-nanoLC (Thermo Scientific, San Jose, CA, USA). Peptides were trapped on 2 cm × 75 µm columns (Thermo Scientific, San Jose, CA, USA). The analytical separation was run for 60 min using a gradient of 99.9% H2O/0.1% (solvent A) and 99.9% ACN/0.1% FA (solvent B) at a flow rate of 300 nl min^−1^. For MS survey scans, the OT resolution was set to 140,000 and the ion population was set to 3 × 10^6^ with an m/z window from 350 to 2000. Twenty precursor ions were selected for higher-energy collisional dissociation (HCD) with a resolution of 35,000, an ion population set to 1 × 10^5^ (isolation window of 0.5 m/z) and a normalized collision energy set to 30%.

#### 4.10.3. Protein Identification and Quantification

Raw data were loaded on Proteome Discoverer 2.2 software for identification and/or quantification of peptides and proteins. Identification was performed in the UniProt/SwissProt (SwissProt human database, containing 20,415 entries) database using Mascot (Version 2.5.1, Matrix Sciences, London). Carbamidomethylation of cysteines, tandem mass tag—sixplex amino terminus and tandem mass tag—sixplex lysine (for labeled samples) were set as fixed modifications and methionine oxidation as a variable modification. Trypsin was selected as the enzyme, with two potential miscleavages. Peptide and fragment ion tolerances were, respectively, 10 ppm and 0.02 Da. The threshold of the average reporter signal-to-noise ratio (S/N) was set to 2. The false discovery ratio (FDR) was set to 1% at peptide-spectrum match (PSM), peptide and protein levels. Only high-confidence master proteins with at least two distinct peptide sequences were required for identifications. All obtained data were tested for significant differences using a two-way ANOVA test in conjunction with Tukey’s multiple comparison test. *p*-values below 0.05 were considered significant.

### 4.11. Metabolomics

Metabolomics studies were conducted based on a previously developed multi-platform methodology [74]. Chromatography was performed on a Waters H-Class Acquity UPLC system composed of a quaternary pump, a column manager and an FTN autosampler (Waters Corporation, Milford, MA, USA). For RPLC analyses, samples were separated on a Kinetex C18 column (150 × 2.1 mm, 1.7 μm) and the corresponding SecurityGuard Ultra precolumn and holder (Phenomenex, Torrance, CA, USA). Solvent A was H_2_O and solvent B was MeCN, both containing 0.1% formic acid. The column temperature and flow rate were set at 30 °C and 300 µL min^−1^, respectively. The gradient elution was as follows: 2% to 100% B in 14 min, hold for 3 min, then back to 2% B in 0.1 min and re-equilibration of the column for 7.9 min. Amide HILIC separations (aHILIC) were conducted on a Waters Acquity BEH Amide column (150 × 2.1 mm, 1.7 µm) bearing an adequate VanGuard precolumn. Solvent A was H_2_O:MeCN (5:95, *v*/*v*) and solvent B was H2O:MeCN (70:30, *v*/*v*) containing 10 mM ammonium formate (pH = 6.5 in the aqueous component). The following gradient was applied: 0% B for 2 min, increased to 70% B over 18 min, held for 3 min, and then returned to 0% B in 1 min and to re-equilibrate the column for 7 min (total run time was 31 min). The flow rate was 500 µL min^−1^, and the column temperature was kept at 40 °C. For the zwitterionic HILIC (zHILIC) method, separation was performed on a Merck SeQuant Zic-pHILIC column (150 × 2.1 mm, 5 μm) and the appropriate guard kit. The following gradient of mobile phase A (MeCN) and mobile phase B (2.8 mM ammonium formate adjusted to pH 9.00) was applied: 5% B for 1 min, increased to 51% B over 9 min, held for 3 min at 51% B and then returned to 5% B for 0.1 min before re-equilibrating the column for 6.9 min (total run time was 20 min) at a flow rate of 300 µL min^−1^ and a column temperature of 40 °C.

In all cases, a sample volume of 10 µL was injected. Samples were randomized for injection, and QC pools were analyzed every six samples to monitor the performance of the analytical platform [75]. 

The UPLC system was coupled to a maXis 3G Q-TOF high-resolution mass spectrometer from Bruker (Bruker Daltonik GmbH, Bremen, Germany) through an electrospray interface (ESI). The instrument was operated in TOF mode (no fragmentation). The capillary voltage was set at −4.7 kV for ESI+, drying gas temperature was 225 °C, drying gas flow rate was set at 5.50 (RPLC), 8.00 (aHILIC) or 7.00 (zHILIC) l min^−1^ and nebulizing gas pressure was 1.8 (RPLC) or 2.0 bar (HILIC). Transfer time was set at 40 (RPLC) or 60 (HILIC) µs and prepulse storage duration at 7.0 (RP) or 5.0 μs (HILIC). For ESI– operation, the capillary voltage was set at 2.8 kV. All the remaining ion source and ion optics parameters remained as in ESI+. Data between 50 and 1000 m/z were acquired in profile mode at a rate of 2 Hz. ESI and MS parameters were optimized using a mix of representative standards fed by a syringe pump and mixed with the LC eluent (mid-gradient conditions) within a tee-junction. Formate adducts in the 90–1247 m/z range were employed for in-run automatic calibration using the quadratic plus high-precision calibration algorithm provided by the instrument’s manufacturer. MS and UPLC control and data acquisition were performed through the HyStar v3.2 SR2 software (Bruker Daltonik, Bremen, Germany) running the Waters Acquity UPLC v.1.5 plug-in.

Run alignment, peak piking and sample normalization were performed on Progenesis QI v2.3 (Nonlinear Dynamics, Waters, Newcastle upon Tyne, UK) and peaks were identified by matching their retention times, accurate masses and isotopic patterns to those of a library of chemical standards (MSMLS, Sigma-Aldrich, Buchs, Switzerland) analyzed under the same experimental conditions, as described elsewhere [74]. A total of 217 compounds were identified. 

The matrix of features was submitted to noise filtering, analytical drift correction (locally weighted scatterplot smoother, LOESS) and sample amount normalization (probabilistic quotient normalization, PQN) by using SUPreMe, an in-house developed software for metabolomics data pretreatment. Multivariate analysis of the resulting data was conducted in SIMCA 15 64 bit (Umetrics, Sartorius Stedim Data Analytics AB, Umeå, Sweden). Volcano plots were created in the MATLAB environment (version R2018a, 64 bit, The Mathworks Inc, Natick, MA, USA) using 2-fold change and 0.05 *p*-value thresholds.

### 4.12. Pathway Enrichment

Proteins and metabolites with a fold change above 1.2 or below −1.2, and a ratio *p*-value < 0.05, were kept for pathway enrichment analysis with MetaCore^TM^ v.6.26 (Clarivate Analytics, London, UK) [76].

## Figures and Tables

**Figure 1 ijms-22-04065-f001:**
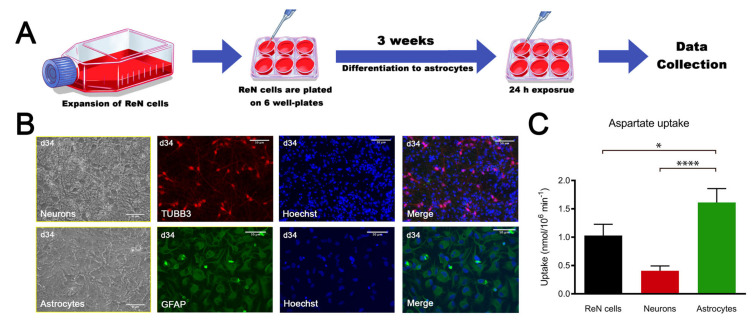
ReN cell-derived astrocytes are functional. (**A**) Scheme of the experiment. (**B**) Bright field pictures from neurons and astrocytes and immunostaining for TUBB3 (red) and glial fibrillary acidic protein (GFAP) (green); nuclei are stained with Hoechst (blue). (**C**) Aspartate uptake of ReN cells, neurons and astrocytes. Results are expressed as mean ± SD, *n* = 10 for ReN cells, *n* = 9 for neurons, *n* = 14 for astrocytes, obtained in 3 independent experiments. Kruskal–Wallis test was followed by Dunn’s multiple comparisons test. * *p* < 0.05, **** *p* < 0.0001.

**Figure 2 ijms-22-04065-f002:**
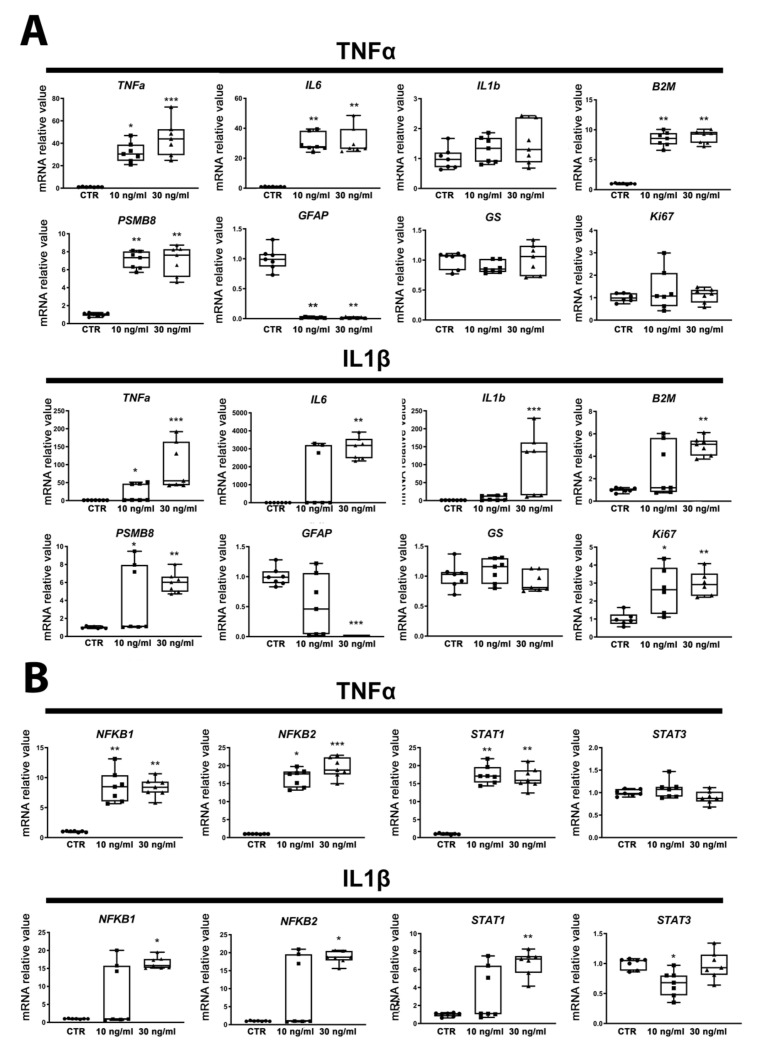
Exposure to cytokines induces human ReN-derived astrocyte reactivity. (**A**) Relative mRNA levels of genes involved in astrocyte reactivity after 24 h of exposure to TNFα or to IL1β. (**B**) Gene expression regulation of transcription factors after 24 h of exposure to TNFα or to IL1β. Results are displayed as boxplots with data points, for each group *n* = 7 samples obtained in 2 independent experiments. The line in the box indicates the median, whereas top and bottom of the box represent the 75th and 25th percentiles; whiskers extend from minimum to maximum values. Statistical analysis was performed by using Kruskal–Wallis test followed by Dunn’s multiple comparisons test. * *p* < 0.05; ** *p* < 0.01; *** *p* < 0.001.

**Figure 3 ijms-22-04065-f003:**
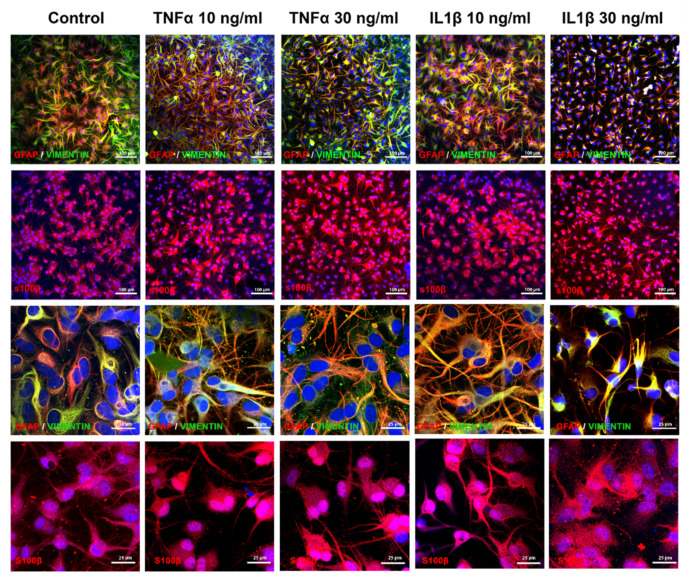
Exposure to cytokines induces morphological modifications in human ReN-derived astrocytes. Double immunostaining for GFAP and vimentin and immunostaining for S100B in control and after 24 h of exposure to TNFα or IL1β. First and second rows: low magnification; third and fourth rows: high magnification.

**Figure 4 ijms-22-04065-f004:**
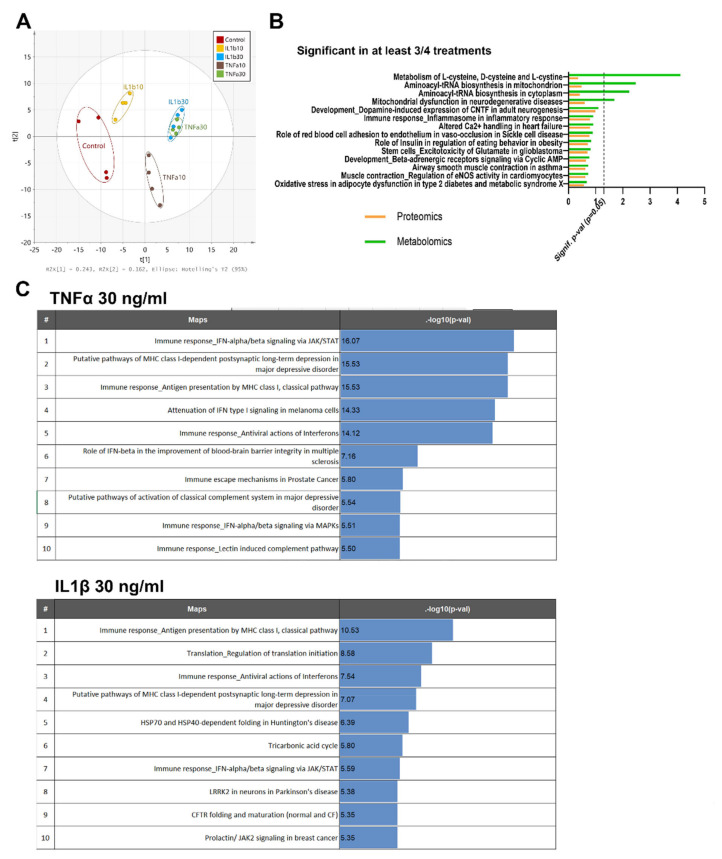
Protein and metabolite pathway analysis after exposure to cytokines. (**A**) PCA of the metabolites measured by three LC–MS approaches, changing after exposure to TNFα or IL1β (10 and 30 ng/mL) for 24 h. (**B**) Main pathways enriched in at least 3 out of 4 treatments after pathway analysis with MetaCore^TM^ v.6.26 (green: metabolomics, orange: proteomics). (**C**) Top regulated protein pathways found after 24 h of exposure to TNFα (30 ng/mL) or IL1β (30 ng/mL). Proteins and metabolites with fold change >1.2 or <−1.2, and a ratio *p*-value < 0.05 were kept for pathway enrichment analysis.

**Figure 5 ijms-22-04065-f005:**
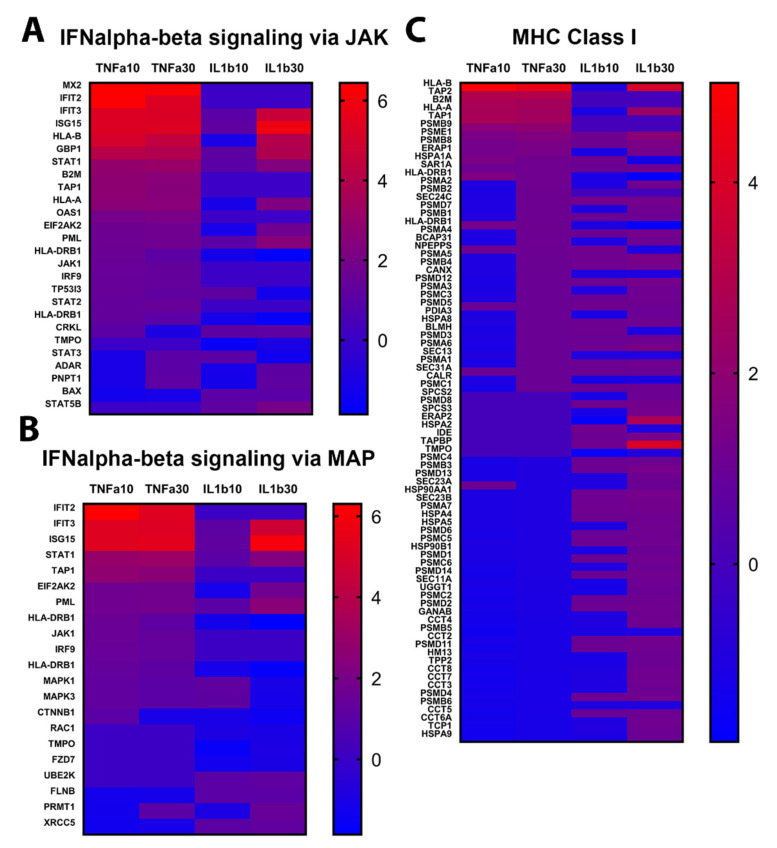
Protein enrichment of inflammatory pathways. Abundance of proteins significantly changed after 24 h of exposure to TNFα or IL1β (10 and 30 ng/mL), as compared to control cultures. (**A**) IFN signaling via JAK pathway. (**B**) IFN signaling via MAP pathway. (**C**) MHC class I pathway. Red: upregulated, blue: downregulated. Scales indicate fold change compared to control cultures.

**Figure 6 ijms-22-04065-f006:**
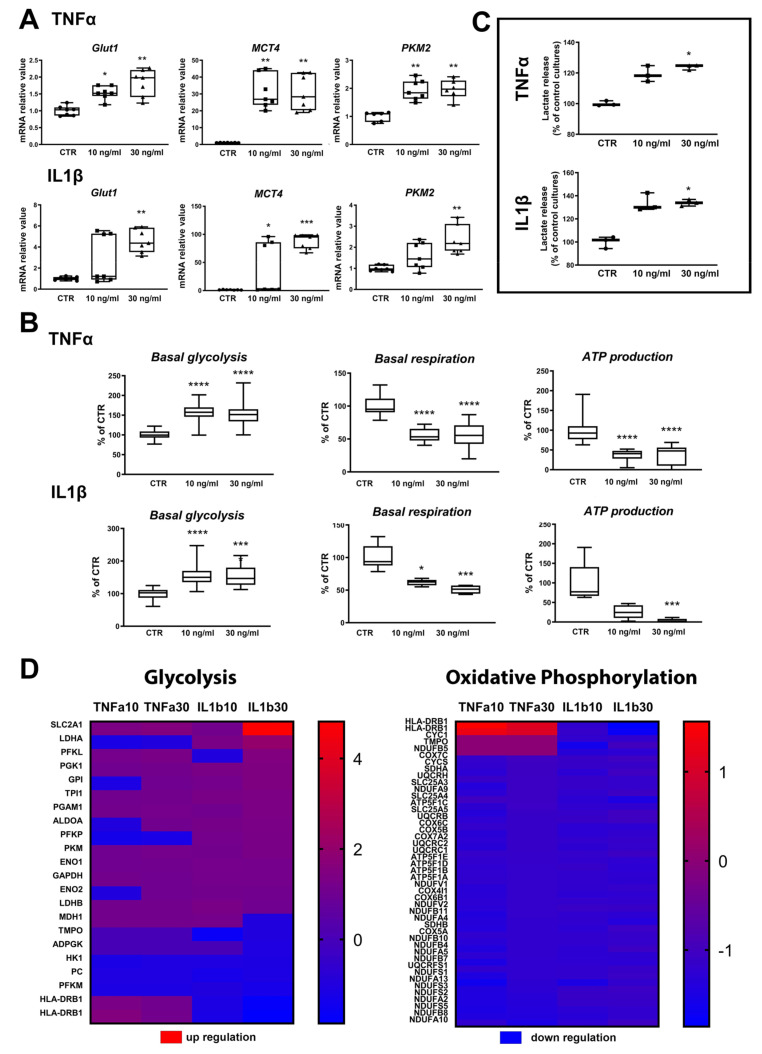
Exposure to cytokines leads to increased glycolysis in ReN-derived astrocytes. (**A**) Changes in genes related to energy metabolism after 24 h of exposure to TNFα or to IL1β. (**B**) Basal glycolysis rate after TNFα (*n* = 48 per group) or IL1β (*n* = 42 per group); basal respiration after TNFα (control: *n* = 18, 10 ng/mL: *n* = 14, 30 ng/mL: *n* = 13) or IL1β (control: *n* = 19, 10 ng/mL: *n* = 5, 30 ng/mL: *n* = 6); and ATP production after TNFα (control: *n* = 17, 10 ng/mL: *n* = 14, 30 ng/mL: *n* = 13); and after IL1β (control: *n* = 8, 10 ng/mL: *n* = 5, 30 ng/mL: *n* = 6); (**C**) Lactate release after 24 h of exposure to TNFα or IL1β. *n* = 3 per group. Results are displayed as boxplots with data points. The line in the box indicates the median, whereas top and bottom of the box represent the 75th and 25th percentiles; whiskers extend from minimum to maximum values. Statistical analysis was performed by using Kruskal–Wallis test followed by Dunn’s multiple comparisons test. * *p* < 0.05; ** *p* < 0.01; *** *p* < 0.001; **** *p* < 0.0001. (**D**) Abundance of proteins significantly changed after 24 h of exposure to TNFα or IL1β as compared to control cultures in glycolysis and oxidative phosphorylation pathways. Red: upregulated, blue: downregulated. Scales indicate fold change compared to control cultures.

## Data Availability

The data that support the findings of this study are available from the corresponding author upon request.

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
