# Peer review of "Neuroinflammatory Response to TNFα and IL1β Cytokines Is Accompanied by an Increase in Glycolysis in Human Astrocytes In Vitro"

_ijms, 2021, doi:10.3390/ijms22084065_

Round 1
Reviewer 1 Report
This manuscript describes for the first time increased glycolysis during human astrocyte activation. Overall, the paper is well written, the results clearly presented and discussed.
Some specific points to address are below:
- The author states that statistical analysis was performed by ANOVA, however, the author didn’t mention if the data met assumptions of ANOVA, i.e., normally distributed data, equal variance among groups. I would highly recommend the author to check the data distribution to justify the use of ANOVA. Non-parametric tests, such as the Mann-Whitney U test, Wilcoxon rank sum test and Kruskal Wallis test, may sometimes be preferable when sample sizes are too small to determine the data distribution.
- Line 61, CNS was not explained the first time it was cited.
- Line 132, the author states: “Although an upregulation of the intermediate filament protein GFAP, has been reported by many authors,…”, please add some references to support this statement.
- Line 135, 137 and 375, the author made several claims based on data that is “not shown”, any particular reason why those data are excluded? I would suggest the author to include all relevant data either in the main text or supplementary files.
- Line 186, MetaCore is a software. I would suggest authors to add the trademark or service mark symbols ( ™, ®, or ℠) to indicate that this name is a trademark the first time it was mentioned.
- Line 271, “in order to functionally evaluate ….SeahorseTM”, please consider rephrasing as this sentence is not easy to follow.
- Line 279, extra comma before increase?
Reviewer 2 Report
This article by Pamies et al. presented results from an in vitro study that characterised the activation of astrocytes (differentiated from human ReN cell line) in response to inflammatory insults with TNF-α and IL-1ß using immunostaining, gene expression, proteomics and metabolomic profiling. For the first time, it showed evidence that astrocyte activation was accompanied by increased glycolysis. The authors showed activation of astrocytes in somewhat divergent manner to classical astrocyte activation (such as decreased GFAP expression, unchanged glutamine synthase expression) and towards a detrimental phenotype (pro-inflmmatory). However, contrary to animal studies, there was an increase in lactate release and MCT4 level which could be either neuroprotective or deleterious that need to be resolved in future studies. This study has great importance to further our understanding of astrogliosis associated with CNS neuroinflammatory diseases.
I have thoroughly enjoyed reading the whole manuscript which is very well written, methodologically sound, have a logical flow of ideas presented, and gave relevant background and interpretation of the results with supporting literature.
I only have some minor comments which the authors might wish to respond-
- please rewrite the sentence in line 77-80 which is almost same as in abstract.
- is it known what percentages of ReN cells differentiates into neurons when cultured in a medium supporting astrocyte differentiation?
- Discussion- line 305-306, I do not think decreased GFAP expression should be considered as a marker of astrocyte activation as there is not many evidence in literature. There might me confounding factors in this study behind reduced GFAP expression. Maybe better to tone down this statement.
